# Integrated Multi-Omics Profiling of Young Breast Cancer Patients Reveals a Correlation between Galactose Metabolism Pathway and Poor Disease-Free Survival

**DOI:** 10.3390/cancers15184637

**Published:** 2023-09-19

**Authors:** Xiangchen Han, Boyue Han, Hong Luo, Hong Ling, Xin Hu

**Affiliations:** 1Department of Breast Surgery, Fudan University Shanghai Cancer Center, Shanghai 200032, China; 19211230033@fudan.edu.cn (X.H.); byhan14@aliyun.com (B.H.); 2Key Laboratory of Breast Cancer in Shanghai, Department of Oncology, Shanghai Medical College, Fudan University, Shanghai 200032, China; 3Precision Cancer Medical Center Affiliated to Fudan University Shanghai Cancer Center, Shanghai 200032, China; luohong0216@163.com

**Keywords:** breast cancer, young patients, genomic, disease-free survival, galactose metabolism, stemness

## Abstract

**Simple Summary:**

In recent years, there has been a notable rise in the incidence of breast cancer among young patients. These young patients exhibit worse survival outcomes and distinct characteristics compared to intermediate and elderly patients. To address this issue, our study focuses on discerning the clinicopathological, genomics, transcriptional features unique to breast cancer in young patients. The ultimate goal is to explore potential therapeutic strategies that can enhance survival outcomes for this particular demographic. Notably, our work identified the novel correlations between galactose metabolism pathways, stemness, and survival outcomes, providing potential therapeutic interventions for young patients.

**Abstract:**

In recent years, there has been a notable rise in the incidence of breast cancer among young patients, who exhibit worse survival outcomes and distinct characteristics compared to intermediate and elderly patients. Therefore, it is imperative to identify the specific features unique to young patients, which could offer insights into potential therapeutic strategies and improving survival outcomes. In our study, we performed an integrative analysis of bulk transcriptional and genomic data from extensive clinical cohorts to identify the prognostic factotrs. Additionally, we analyzed the single-cell transcriptional data and conducted in vitro experiments. Our work confirmed that young patients exhibited higher grading, worse disease-free survival (DFS), a higher frequency of mutations in TP53 and BRCA1, a lower frequency of mutations in PIK3CA, and upregulation of eight metabolic pathways. Notably, the galactose metabolism pathway showed upregulation in young patients and was associated with worse DFS. Further analysis and experiments indicated that the galactose metabolism pathway may regulate the stemness of cancer cells and ultimately contribute to worse survival outcomes. In summary, our finding identified distinct clinicopathological, transcriptional, and genomics features and revealed a correlation between the galactose metabolism pathway, stemness, and poor disease-free survival of breast cancer in young patients.

## 1. Introduction

Breast cancer (BC), constituting 31% of female cancers, is the most commonly diagnosed malignancy and the second leading cause of morbidity among women worldwide [1]. In particular, breast cancer in young individuals has worse histological grading, higher ki-67, and more lymphocyte node infiltration, leading to more relapse and poorer survival outcomes [2,3]. Recent years have witnessed a gradual rise in the incidence of breast cancer among young individuals, necessitating focused attention in both research endeavors and clinical practice [4].

Notably, previous studies have consistently demonstrated that young patients present with more severe clinicopathological features and unfavorable prognosis [5]. However, the underlying biological mechanisms driving these disparities have remained elusive. Tumors, including breast cancer, are widely acknowledged to exhibit significant intertumoral and intratumoral heterogeneity [6]. By leveraging transcriptional and genomic profiling, we sought to elucidate the heterogeneity between young and elderly BC patients and gain a deeper understanding of the biological basis of the observed differences between these two patient groups. It is widely recognized that tumors consist of a heterogeneous mix of cells, and the application of single-cell RNA-sequencing (scRNA-seq) on isolated tumor tissue allows for the characterization of heterogeneous tumor cells [7,8]. Notably, scRNA-seq enables the molecular distinction of identical cell types within a complex population mix, thereby revealing the distinct characteristics of specific cell types.

The mounting evidence highlights the involvement of metabolic regulation in cancer progression, metastasis, and therapy resistance [9,10]. Therefore, understanding metabolic heterogeneity among patients becomes crucial in predicting diverse survival outcomes. Our previous study has successfully validated the metabolic heterogeneity in triple negative breast cancer (TNBC) patients and proposed distinct clinical characteristics and potential therapeutic approaches for each subtype [11,12,13]. Building upon these discoveries, the recent study aims to investigate the metabolic difference between different age groups, thereby elucidating the underlying factors contributing to the diverse prognosis observed.

Tumor stemness can be described as the characteristics and properties exhibited by stem cells, which are associated with tumor initiation, progression, metastasis, therapy resistance, and relapse [14,15,16,17,18]. The expression of breast cancer stem cell (CSC) markers, including CD44, CD29, and CD49f, is correlated with aggressive tumor features and an increased risk of recurrence.

Interestingly, our work revealed the upregulation of the galactose metabolism pathway specifically in young patients through the bulk RNA-seq data, and further uncovered a correlation between galactose metabolism and stemness through scRNA-seq data. Moreover, we experimentally validate this correlation in cell lines. Our work provides a novel perspective to explore the underlying biological basis of poor prognosis in young BC patients.

## 2. Material and Methods

### 2.1. Study Cohorts and Statistical Analyses

Both clinical, transcriptional, and genomic data from the Molecular Taxonomy of Breast Cancer International Consortium (METABRIC) cohort were sourced from the cBio-Portal for Cancer Genomics [19]. A total of 1980 patients were encompassed. Clinical data for 4227 breast cancer patients were obtained from the Fudan University Shanghai Cancer Center (FUSCC). Furthermore, the Memorial Sloan Kettering Cancer Center (MSKCC) cohort contributed genomic data for 807 primary breast cancer patients [11], and the Swedish Cancerome Analysis Network—Breast (SCAN-B) cohort contributed transcriptional data for 3069 patients. The entire workflow of the study is visually presented in Figure 1.

To analyze the somatic mutation profiles of different patient groups, the maftools package (v2.6.05) was employed. The primary endpoints of interest were overall survival (OS) and disease-free survival (DFS). Kaplan–Meier plots were used to estimate the survival rates among different age groups, and the log-rank test was utilized to identify the metabolic pathways associated with DFS. All bioinformatics analyses were performed using R version 4.2.0 and Python version 3.9.

### 2.2. scRNA-Seq Cohorts and Data Processing

The quality-controlled scRNA-seq data of BC patients were deposited at Gene Expression Omnibus (GEO accession numbers: GSE176078) and 31 treatment-naïve patients were enrolled for subsequent analysis [7]. Briefly, data normalization, batch correction, dimensionality reduction, and cell clustering were performed via the Seurat4 package (v4.1.0). Later, cell clusters were annotated using the Garnett package (v0.2.18) and inferCNV package (v1.6.0). The normalized enrichment score (NES) of the metabolic pathways for each cell was calculated using the ssMWW-GST function [20]. Ultimately, the cytoTRACE score for individual cells was calculated via the CYTOTRACE package (v0.3.3) [21].

### 2.3. Cell Lines and D-Galactose Treatment

Human BC cell lines, including Hs578t, MDA-MB-231, MDA-MB-468, BT549, HCC1806, BT20, and HCC1937 cell lines, were purchased from the American Type Culture Collection (ATCC) and confirmed without mycoplasma contamination. Hs578t and MDA-MB-231 cells were cultured in high-glucose DMEM (BasalMedia, Shanghai, China, L110) supplemented with 10% fetal bovine serum (FBS, Gibco) and 1% penicillin-streptomycin (BasalMedia, S110B) at 37 °C with a 5% CO_2_ incubator.

For the D-galactose treatment, cells were seeded in six-well plates and allowed to incubate overnight. Subsequently, 25 mM D-galactose (Sigma-Aldrich, St. Louis, MI, USA, G5388) was added to the growth medium [22]. Cells were cultured for 24 h for RNA-extraction experiments and 120 h for the mammosphere formation assay.

### 2.4. RNA Extraction, RT, and Real-Time qPCR

Total RNA was extracted using the RNA-Quick Purification Kit (ES science, RN001), and reverse-transcribed (RT) into complementary DNA (cDNA) using the HiScript II 1st Strand cDNA Synthesis Kit (Vazyme Biotech, Nanjing, China, R211). Real-time quantitative Polymerase Chain Reaction (qPCR) was performed using the ChamQ Universal SYBR qPCR Master Mix (Vazyme Biotech, Q711). All the primers for real-time qPCR are provided in Appendix A.

### 2.5. Mammosphere Formation Assay

The mammosphere culture was conducted following the procedures described in a prior study [17]. MDA-MB-231 and Hs578t cells were seeded in 24-well ultra-low attachment plates (Corning, Corning, NY, USA, 3473) at a density of 2 × 10^3^ cells per well. Cells were cultured in DMEM/F12 reduced serum medium (Gibco, Billings, MT, USA, 12364010) supplemented with B27 (ThermoFisher, Waltham, MA, USA, 17504044), 20 ng/mL epidermal growth factor (EGF, Peprotech, Cranbury, NJ, USA, 100-47), 20 ng/mL basic fibroblast growth factor (bFGF, ThermoFisher, 13256029), 0.4% albumin from bovine serum (BSA, BasalMedia, S476T7), 2 μg/mL heparin (STEMCELL, Vancouver, BC, Canada, 07980), and insulin-transferrin-selenium (ITS-G, BasalMedia, S450J7). Following a 5-day incubation period, the quantification of mammospheres (diameter > 50 μm) was performed, and representative images were captured.

## 3. Results

### 3.1. Clinicopathological and Survival Features of BC in Young Patients

From the METABRIC cohort, 1980 patients with BC were eligible for further analyses. These patients were categorized into different age groups: the young group comprising 120 patients (aged ≤39 years), the intermediate group consisting of 755 patients (aged 40 to 59 years), and the elderly group comprising 1105 patients (aged ≥60 years), as depicted in Figure 2B. Otherwise, the FUSCC cohort exhibited a larger proportion of patients in the young and intermediate groups compared with the METABRIC cohort (Figure 2A and Appendix A). Clinicopathological features of individual groups were analyzed, and significantly different features (*p* < 0.001) between groups are illustrated in Figure 2C. In the METABRIC cohort, young patients displayed higher histological grading and worse tumor stage. Additionally, young patients are more likely to be negative for estrogen receptors (ER) and progesterone receptors (PR), while positive for human epidermal growth factor receptor 2 (HER2).

We further investigated the survival outcomes of patients within different groups. The young group showed worse OS (overall survival) and DFS (disease-free survival) compared to both the intermediate group (OS *p* = 0.050; DFS, *p* = 0.002) and the elderly group (OS *p* = 0.003; DFS, *p* = 0.008) (Figure 2D,E). Likewise, the young group demonstrated worse DFS in the FUSCC cohort (Appendix A).

### 3.2. Genomics Features of BC in Young Patients

Given that the mutation of oncogenes and tumor suppressor genes plays a crucial role in tumor progression, we explored the genomic heterogeneity of patients across various age groups in the METABRIC cohort (including 86 patients with no detected mutations). The young group exhibited a higher frequency of mutations in TP53 (young vs. intermediate vs. elderly: 61.7% vs. 40.8% vs. 29.0%, *p* < 0.001), BRCA1 (7.8% vs. 1.6% vs. 1.6%, *p* < 0.001), and COL22A1(8.7% vs. 3.5% vs. 4.1%, *p* = 0.019), and HDAC9(3.4% vs. 0.8% vs. 0.8%, *p* = 0.031); a lower frequency in PIK3CA (22.6% vs. 42.0% vs. 45.1%, *p* < 0.001) and KMT2C (4.3% vs. 10.5% vs. 14.4% *p* = 0.001) (Figure 3A,B). Furthermore, young patients had a higher prevalence of somatic mutations in the TP53 signaling family (62% vs. 42% vs. 29%, *p* < 0.001) and a lower frequency in the PI3K signaling family (30% vs. 49% vs. 55%, *p* < 0.001) (Figure 3C). To further explore the genomic features of the young patients, we found co-occurrent mutations in PIK3CA and AHNAK2 and mutually exclusive mutations in TP53 and ANNAK (Figure 3D). Remarkably, similar trends were observed in the MSKCC cohorts (Appendix A).

### 3.3. Transcriptomic Features of BC in Young Patients

To investigate the potential mechanisms contributing to poorer survival outcomes in the young group, we performed the Mann–Whitney–Wilcoxon gene set test (MWW-GST) and calculated the enrichment normalized score (NES) of 65 metabolic pathways in individual bulk RNA-seq samples from the METABRIC cohort. Notably, the young group exhibited significant upregulation of eight pathways (logFC (NES) > 0.1 and qValue < 0.01) compared with the intermediate and elderly groups (Figure 4A), including the glycosphingolipid biosynthesis pathway. Moreover, 12 pathways (HR >1 & *p* < 0.01) were identified as determinant prognostic features for worse DFS (Figure 4B), such as the fructose and mannose metabolism pathway (HR 1.58, 95% CI 1.300–1.930, *p* < 0.001, downregulation as reference). Among these metabolic pathways, six pathways demonstrated simultaneous upregulation in the young group and statistical significance for DFS via multivariate analysis (Figure 4C,D), including the galactose metabolism pathway. Remarkably, consistent findings were observed in the SCAN-B cohort (Appendix A).

### 3.4. Galactose Metabolic Pathways Are Upregulated in Less-Differentiated Cancer Cells

To gain insight into how galactose metabolism may contribute to worse DFS, we focused our analysis on cells from 31 treatment-naïve breast cancer samples in the published single-cell dataset (GSE176078) (Figure 5A and Appendix A). Following quality control, data integration, batch-effect removal, and cell annotation, 28,661 cancer cells were identified and re-clustered into 16 clusters (Figure 5B). We then assessed the NES of galactose metabolism pathways in individual cancer cells, and cells in cluster 1, 5, 6, 7, 14, and 15 showed upregulation of the metabolism pathways and were annotated as HGM (hyper-galactose metabolic) cells (Figure 5C,D). Additionally, HGM cells significantly expressed UGP2, HK1, and PFKL, which are critical genes in the galactose metabolism pathway (Figure 5E).

To uncover the characteristics of HGM cells, we calculated the cytoTRACE score of individual cells, which is indicative of their differentiation states. Interestingly, HGM cells displayed significantly higher cytoTRACE scores compared to others (Figure 5F,G), indicating their less-differentiated status. Previous studies have associated less-differentiated cells with relapse and metastasis. Among all cancer cells, the cytoTRACE score positively correlated with the NES of galactose metabolism (R = 0.47 *p* < 0.001) (Figure 5H). Additionally, the cytoTRACE score exhibited significant correlations with the expression of critical genes in the galactose metabolism pathway, including UGP2, HK1, PFKL, GLB1, GALK1, GAA, G6PC3 and B4GALT1 (R > 0.3 *p* < 0.001) (Figure 5I).

### 3.5. Galactose Metabolism Regulates Breast Cancer Stemness

To validate the influence of galactose metabolism in regulating BC stemness in vitro, BC cell lines were utilized. According to the RNA-seq data from 19 cell lines, Hs578t, MDA-MB-231, MDA-MB-436, HCC1395, BT549, HCC1143, and HCC1937 cell lines showed higher enrichment scores of galactose metabolism and stemness signature. Moreover, the CD29^high^CD24^+^ (stem-like) subpopulation was found in these cell lines (Figure 6A). Subsequently, we examined the impact of galactose metabolism on the tumor cells. Interestingly, D-galactose treatment significantly enhanced the mammosphere formation efficacy of MDA-MB-231 and Hs578t cells (Figure 6B,C). Meanwhile, the real-time qPCR results showed the upregulation of GALK1, UGP2, G6PC3, ITGA6, ITGB1, and CD44 in cells treated with D-galactose (Figure 6D,E), suggesting the potential role of galactose metabolism in regulating stemness.

## 4. Discussion

The result of our study confirmed the differences in breast cancer among young patients compared to other age groups. In terms of clinicopathological features, breast cancer in young individuals was characterized by higher histological grading, greater likelihood of being ER/PR/HER2 negative, and worse DFS. Examining the multi-omics perspective, young patients harbored more somatic mutations in TP53, BRCA1, COL22A1, and HDAC9, but fewer mutations in PIK3CA and KMT2C; harbored more mutations in the TP53 signaling family, but fewer in the PI3K-AKT-mTOR signaling family. Additionally, our analysis revealed an upregulation of the galactose metabolism pathway in young patients, which emerged as a significant prognostic feature linked to worse DFS outcomes. Eventually, the scRNA-seq data from the breast cancer patients and experimental results from cell lines confirmed that galactose metabolism showed the potential to modulate cancer stemness, ultimately contributing to the poorer DFS outcomes observed in young breast cancer patients.

We conducted an extensive genomic analysis to elucidate the prognostic mutations and potential therapeutic targets. Notably, young patients displayed a higher incidence of mutations in the BRCA1 and TP53 genes, which were also evident in the corresponding pathways. Emerging evidence indicates that the TP53 mutation is inclined to reach a complete response (CR) after chemotherapy; however, it is also linked to worse OS and DFS in BC patients [23,24,25]. Furthermore, TP53-mutated patients showed a higher risk of tumor recurrence following breast-conserving therapy (BCT) than wild-type patients, thereby warranting a recommendation of mastectomy for TP53-mutated individuals [26]. In contrast, a lower mutation frequency of PIK3CA was observed in young patients, consistent with the findings in the PI3K pathway. A large pooled analysis found that PIK3CA mutation was associated with increasing age and improved DFS and OS in early-stage BC [27]. Additionally, the results from the SAFIR02 trial (NCT02299999) indicated that PIK3CA mutation was associated with a better OS in metastatic triple-negative BC [28]. These genomic features may provide insight into the rationale behind the poorer survival outcomes observed in young BC patients.

From the transcriptional data, we observed the significant upregulation of galactose metabolism pathways in the young patient groups and identified the galactose metabolism pathway as a prognostic factor for worse survival outcomes.

Stemness can be described as the characteristics and properties exhibited by stem cells, which is associated with tumor initiation, progression, metastasis, therapy resistance, and relapse. Devon A. Lawson et al. demonstrated that metastases originate from stem-like cells that proliferate and differentiate, ultimately giving rise to advanced metastatic breast cancer, a process mitigated by treatment with cyclin-dependent kinase (CDK) inhibitors. Our previous study also underscored the significance of interfering with PROCR, identified as a marker of breast cancer stem cells (BCSCs), in reducing BCSC numbers, arresting tumor growth, and preventing rapid tumor recurrence. Furthermore, various treatments targeting cancer stem cells (CSCs) are available for clinical application, with some undergoing clinical trials. These therapeutic strategies encompass the inhibition of pivotal CSC signaling pathways, such as the WNT and NOTCH pathways; ablation utilizing antibodies or antibody–drug conjugates (ADCs), including CD33 antibodies; and epigenetic therapy, featuring histone deacetylase inhibitors (HDACis). Therefore, stemness emerges as a key contributor to breast cancer relapse and a potential therapeutic target.

Notably, recent studies have highlighted the role of galactose metabolic enzymes in modulating the stemness of cancer stem cells (CSC) and influencing clinical outcomes. In the Leloir pathway, galactose is initially converted to galactose-1-phosphate by galactokinase (GALK1), which allows cancer cells to utilize galactose as an alternative fuel source instead of glucose in glioblastoma (GBM) [29]. Also, the expression of GALK1 has been found to be correlated with poor clinical outcomes in GBM. Moreover, GALK1 has also been implicated in promoting high epithelial–mesenchymal transition (EMT) and worse OS in colorectal cancer (CRC) [30]. Additionally, the conversion of glucose-1-phosphate to UDP-glucose, a critical step in galactose metabolism, is catalyzed by UDP-glucose pyrophosphorylase-2 (UGP2). UGP2 has been recognized as a crucial factor in cancer maintenance and has emerged as a potential therapeutic target in pancreatic ductal adenocarcinoma (PDAC) [31]. UGP2 upregulation has been observed in leukemic stem cell-enriched factions and has been identified as a determinant prognostic factor in OS and DFS [32]. Another enzyme, Beta-1,4-Galactosyltransferase 1 (B4GALT1), has been implicated in the maintenance of the stemness in lung adenocarcinoma (LUAD) and CRC [33,34]. Our findings align well with previous studies, collectively suggesting that the galactose metabolism pathway may indeed play a pivotal role in modulating the stemness of BC cells. Therefore, targeting this pathway could hold promise as a potential therapeutic approach to improve survival outcomes in young breast cancer patients.

Tumors consist of a heterogeneous mixture of cells. Gene expression studies using bulk RNA samples only offer an average representation of these diverse cell types. In contrast, single-cell studies empower us to molecularly distinguish between all cell types within this complex mixture. Our results revealed significant heterogeneity in galactose metabolism among the cancer cells. We proceeded to investigate the characteristics of cells exhibiting a high galactose metabolism score, ultimately identifying that they were in a less differentiated state. To delve deeper into the characteristics of cells displaying a high galactose metabolism score, we can identify the differentially expressed genes (DEGs) and identify potential markers, which can then be validated through multiplex immunohistochemistry (mIHC). Additionally, we can predict cell communication patterns between these cells and immune cells, pinpoint specific associations with distinct immune cell subtypes, and subsequently validate these findings through mIHC or immune experiments.

We performed limited in vitro experiments to initially validate our bioinformatics analysis findings. The results of the mammosphere formation efficacy assay and real-time qPCR indicated the potential role of galactose metabolism in regulating stemness. To further strengthen these relationships, we plan to assess galactose metabolism levels using the α-galactosidase (α-GAL) Activity Assay Kit and quantify the protein levels of critical galactosidases in future studies. Additionally, we intend to manipulate galactose levels in the culture medium by adding or eliminating D-galactose and explore the use of galactosidase inhibitors.

The current study has limitations. Firstly, the clinicopathological and multi-omics features identified in young breast cancer patients were based on retrospective cohorts, which may introduce inherent biases and limit the generalizability of our findings. Secondly, the scRNA-seq data available for young patients were insufficient in quantity, hindering a comprehensive analysis of significantly enriched cell clusters in this subgroup. Thirdly, while we have identified the galactose metabolism pathway as a potential prognostic factor in young BC patients, the underlying mechanism by which this pathway modulates stemness remains unexplored. Additionally, further investigation into potential therapeutic targets within this pathway is warranted to enhance its clinical relevance. Lastly, in order to translate our findings into clinical practice, thorough validation in prospective cohorts is essential and rigorous clinical validation is imperative to ascertain their efficacy and safety profiles.

## 5. Conclusions

Through the integration and comprehensive analysis of multi-omics data in breast cancer (BC), we have successfully identified distinct clinicopathological characteristics, specific genetic vulnerabilities, and metabolic features unique to young patient groups. Specifically, our investigation revealed a significant upregulation of the galactose metabolism pathway in these young patients. Furthermore, we unveiled its potential role in modulating stemness, ultimately contributing to the observed worse survival outcomes. These findings hold promising implications for the development of potential therapeutic strategies for young BC patients.

## Figures and Tables

**Figure 1 cancers-15-04637-f001:**
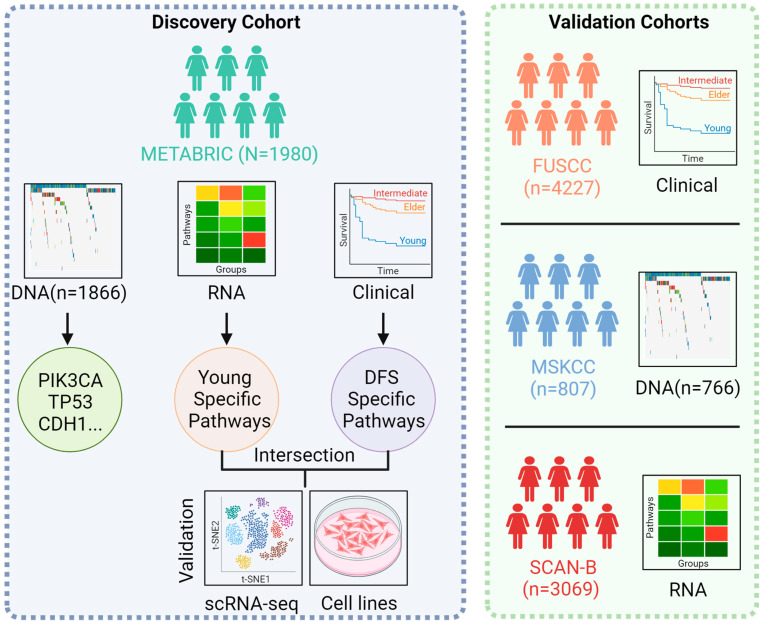
Graphical overview of the investigation.

**Figure 2 cancers-15-04637-f002:**
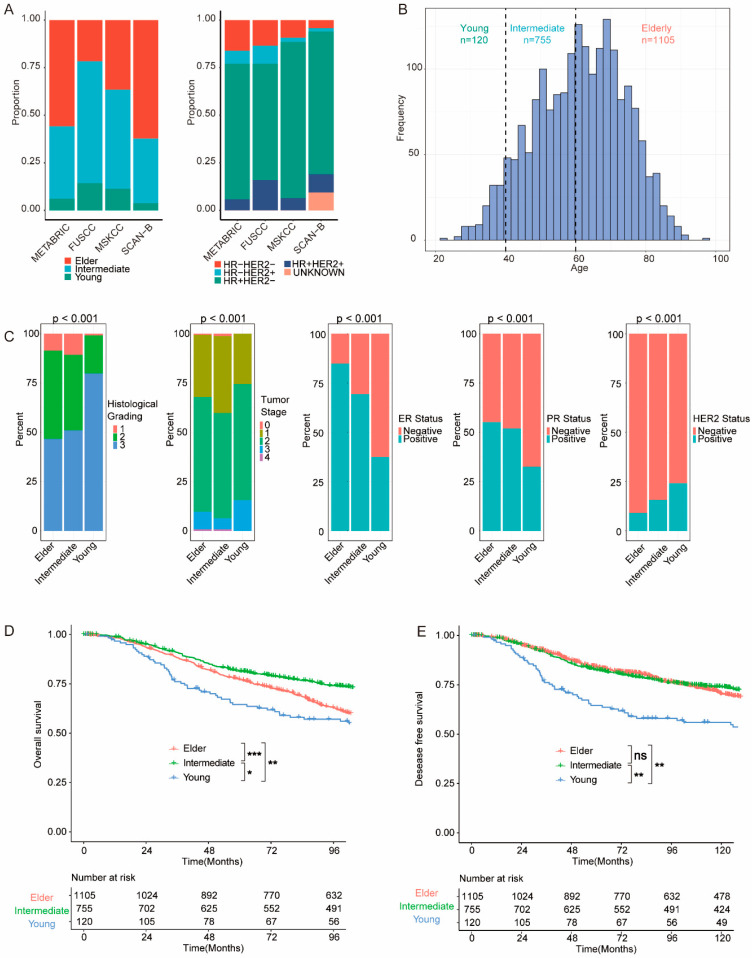
Clinicopathological and survival features of breast cancer patients across different age groups. (**A**) Age group and subtype distribution across the METABRIC, FUSCC, MSKCC, and SCAN-B cohorts. (**B**) Distribution of age at diagnosis and the defined age groups in the METABRIC cohort. (**C**) Histological grading, tumor stage, estrogen receptor (ER) status, progesterone receptor (PR) status, and human epidermal growth factor receptor 2 (HER2) status in each age group. (**D**) Overall survival (OS) in the MTEABRIC cohort. (**E**) Disease-free survival (DFS) in the METABRIC cohort. Survival comparisons were performed, and significant differences were annotated as * *p* < 0.05; ** *p* < 0.01; *** *p* < 0.001; or not significant (ns) (*p* > 0.05).

**Figure 3 cancers-15-04637-f003:**
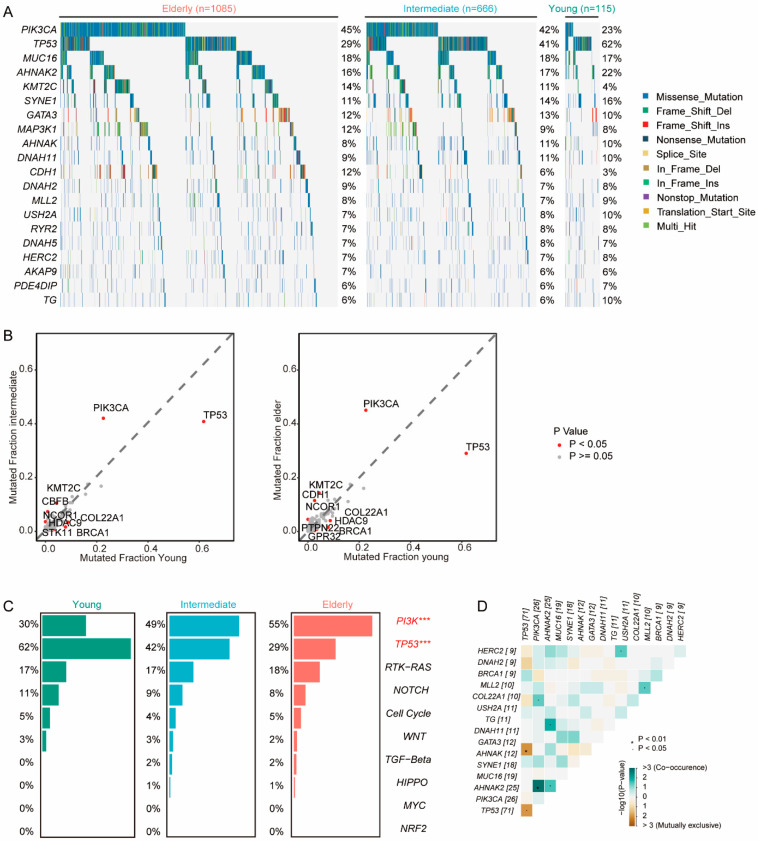
Genomic features of breast cancer patients in each age group. (**A**) Mutation profile of patients in elderly, intermediate, and young groups. The top 20 genes are listed. (**B**) Comparison of mutations in young, intermediate (**left**), and elderly (**right**) groups. (**C**) Mutations of oncogenic pathways in young, intermediate, and elderly groups. *** FDR < 0.001 (**D**) Co-occurrence and mutually exclusive mutations in the young group.

**Figure 4 cancers-15-04637-f004:**
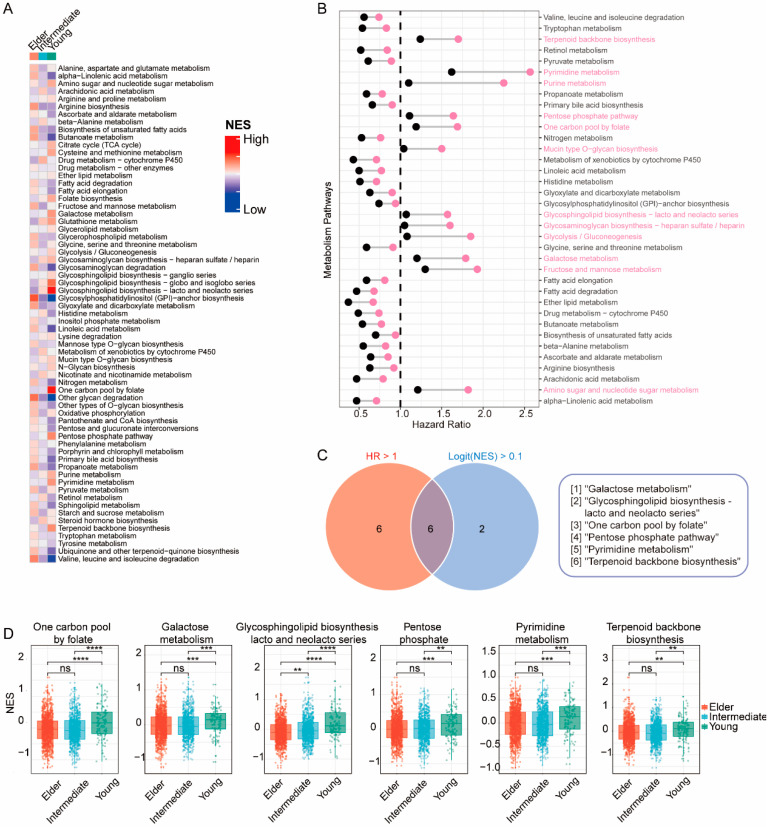
Metabolic heterogeneity of breast cancer patients in different age groups. (**A**) Heatmap displaying the normalized enrichment score (NES) of 65 metabolic pathways in each age group. (**B**) Forest plot illustrating the hazard ratio (HR) and 95% confidence interval (CI) of each metabolic pathway in the METABRIC population. (**C**) Venn diagram of the intersection between metabolic pathways upregulated in the young group (logit (NES) > 0.1) and pathways associated with worse disease-free survival (HR > 1). (**D**) Box plots presenting NES values of intersected metabolic pathways in each age group; *p*-values: Kruskal–Wallis test for multiple comparisons. ** *p* < 0.01, *** *p* < 0.001, **** *p* < 0.0001, and ns *p* > 0.05.

**Figure 5 cancers-15-04637-f005:**
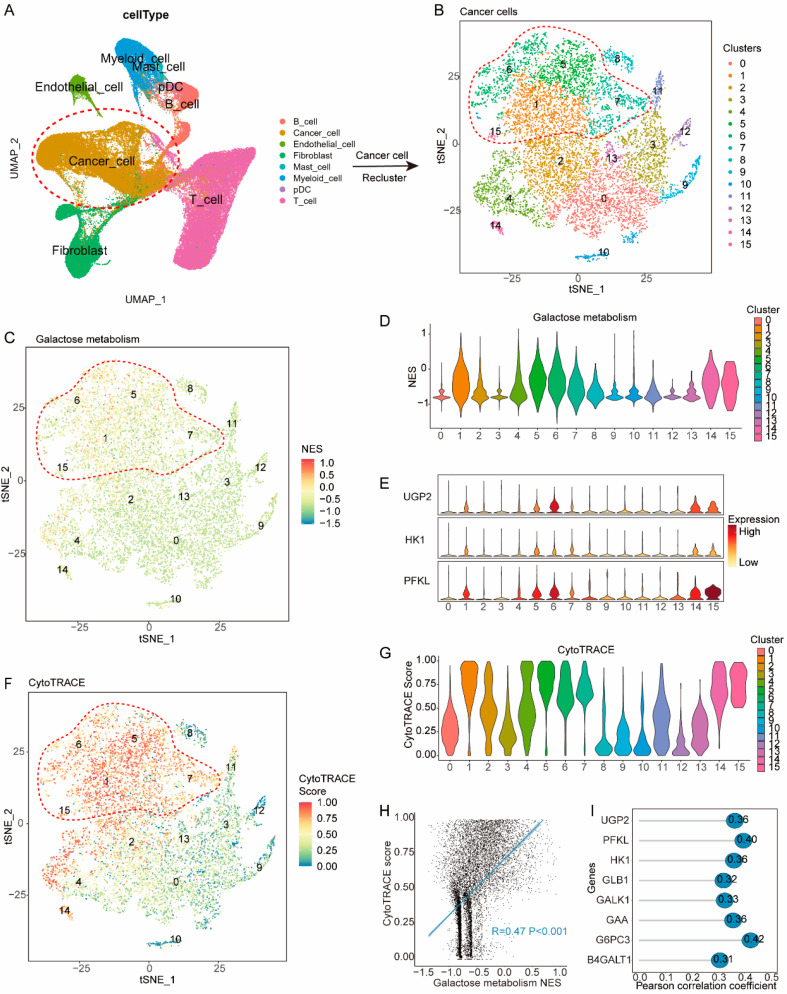
Correlation of the galactose metabolic pathway with less-differentiated cancer cells. (**A**) tSNE plot of 84,854 cells (GSE176078) color-coded to indicate cell type. (**B**) Sub-clustering cancer cells into 16 clusters, as indicated by the color-coded legend. (**C**,**D**) tSNE and violin (**D**) plots showing the NES of the galactose metabolism pathway for each cluster. (**E**) Violin plot showing the expression of UGP2, HK1, and FPKL for each cluster. (**F**,**G**) tSNE (**F**) and violin (**G**) plot showing the CytoTRACE score for each cluster. (**H**) Dot plot showing the NES of galactose metabolism pathway positively correlated with the CytoTRACE score. (**I**) Lollipop plot showing the expression of critical genes in the galactose metabolism pathway positively correlated with the CytoTRACE score.

**Figure 6 cancers-15-04637-f006:**
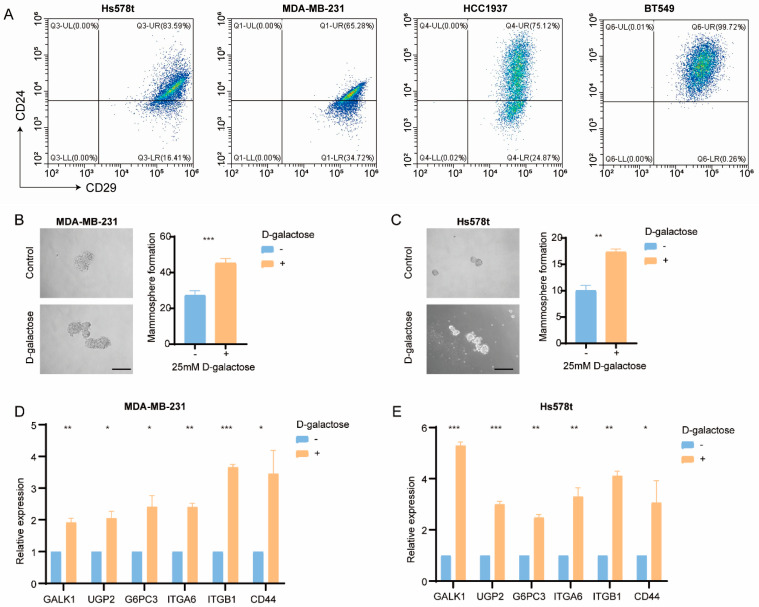
Correlation of the galactose metabolic pathway with stemness in breast cancer cell lines. (**A**) Representative flow cytometry plots of Hs578t, MDA-MB-231, HCC1937, and BT549 cells using the cancer stem cell (CSC) markers CD29 and CDC4. (**B**,**C**) Representative images (**left**) and mammosphere formation counting (**right**) of mammospheres formed from MDA-MB-231 (**B**) and Hs578t (**C**) cells treated with vehicle and D-galactose. Scale bars, 100 µm. (**D**,**E**) qRT-PCR analysis of GALK1, UGP2, G6PC3, ITGA6, ITGB1, and CD44 expression in MDA-MB-231 (**D**) and Hs578t (**E**) cells treated with vehicle and D-galactose. * *p* < 0.05, ** *p* < 0.01, *** *p* < 0.001.

## Data Availability

We obtained clinical, transcriptional, and genomic data for the METABRIC and MSKCCC cohorts from the cBioPortal website (https://www.cbioportal.org/ (accessed on 4 September 2022). Clinical and transcriptional data for the SCAN-B cohort were acquired from the GEO database (GSE96058).

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
