# Peer review of "Integrated Multi-Omics Profiling of Young Breast Cancer Patients Reveals a Correlation between Galactose Metabolism Pathway and Poor Disease-Free Survival"

_cancers, 2023, doi:10.3390/cancers15184637_

Round 1

Reviewer 1 Report

The manuscript by Han and colleagues describes their study into identifying prognostic factors for pre-menopausal breast cancer patient. Initially the authors compiled data from three different breast cancer cohorts and extrapolated transcriptomic and genomic sequencing data. Clinicopathologic analysis demonstrated that patients aged below 40 years of age had higher histological grading, worse tumor stage, higher level of TNBC, and lower DNS and OS than older patients. Genomic analysis demonstrated differential mutation frequencies in younger BC patients than in older BC patients, with the largest changes being lower frequency of PI3KCA and higher frequency of P53 genes. Transcriptionally the authors focused specifically on 65 metabolic pathways with six pathways found to be upregulated in the young BC patient group, including the galactose metabolism pathway. The authors then pivot to analysis a publicly available single-cell dataset from 35 BC patients and demonstrated that activation of galactose metabolism pathway activation correlated with less differentiated cancer cells, leading the authors to investigate if galactose metabolism is correlated with stemness in cancer cells, which was demonstrated in breast cancer cell lines treated with galactose.

Focusing on the galactose metabolism pathway in subsequent studies seems somewhat arbitrary as the results from figure 4 show that other metabolic pathways are quantitatively equally if not more different between the age groups. A better rationale for why there was so much focus on the galactose pathway would be indicated.

For the single-cell analysis the authors mention in the materials and methods that the data comes from 36 patients, but in the results, there is an indication that it comes from 35 patients.

Is there some data on the 35/36 patients age? If these are from across all age groups, there is still validity to the observations, but it would be impossible to link worse disease with stemness increase in young BC patients as suggested in the discussion.

Could the authors elaborate a little more on the findings in figure 6A? Are these data showing the general CD24/CD29 distribution of the cells or are these after treatment? The authors mention that these are FACS data; does that mean the cells were enriched before treatment?

The discussion seems to only address the clinicopathology data, genomic data and transcriptomic data, but there is no discussion of the single-cell analysis or the in-vitro studies.

Reviewer 2 Report

2.1 More explanation for study sample. Very confused the presentation

Good but the english could be improved

Reviewer 3 Report

In this paper, Han et al., try to understand the causes of the worse survival outcomes of young breast cancer patients, compared to the older ones.

Their approach is to integrate multi-omics data from available datasets, in order to identify clinicopathological characteristics and genetic vulnerabilities, unique to the young patient groups.

The article is well written, clear in its initial rationale and architecture.

However, while the topic is very interesting in principle, the experimental approaches suffer from many weaknesses that diminish the enthusiasm for the results and need to be addressed before the paper could be considered for publication.

Major issues:

1)    The authors use the data from all BC subtypes together, making the genetic and transcriptional results much less compelling. They should comment on that and describe the number of patients from each subtype, included in each dataset.

2)    All analyses are made according to 3 age categories (≤39 years; 40 to 59 years; ≥60 years).

How they make this choice is not commented. The value of the intermediate category quite uncertain.

3)    Despite claiming that they analyze the transcriptomic features of BC, the authors decide to focus only the metabolic pathways and then, particularly, of the galactose metabolism pathway. The reason for that is questionable and, however, not discussed. Also, the positive association between galactose metabolism pathway and worse disease-free survival seems quite weak, a Kaplan-Meier plot would be appreciated.

4)    The single cell analyses and the in vitro validation of the galactose metabolism pathway are quite sketchy and superficial. The authors arbitrarily decide what to look for and where. The relevance of the results is doubtful.

Technical/scientific points:

1) Rationale for the study: Why do they focus on metabolism? Why the galactose route? Why stemness? (On this last point there is only one sentence in the discussion). The previous works on young patients should perhaps be discussed and commented on whether the results on the mutations are in line with theirs.

2) In “Materials and Methods” they indicate to use clinical, transcriptional and genomic data from 3 cohorts (METABRIC, FUSCC and MSKCC) integrating the information as described in Fig.1. We expected the data to be more integrated or to show one cohort in the main figure and the other two in the supplementary ones, but this is not the case. Mostly the results are on the METABRIC cohort, something on FUSCC and only one figure (Supp 2) on the MSKCC cohort.

3) Fig. 1D: They report in the text a higher significance in overall survival between Y and Elder, while in Figure the significance is clearly higher between Y and Intermediate (**), lower between Y and Elder (*).

4) They do not describe exhaustively how they choose patients. In some analyses (Fig 3A, Supp 2A) the sum of the number of patients (divided by age group) does not correspond to the number of patients indicated in M&M for that cohort. Were any patients excluded from the analysis? If yes, how and why? Even the captions are missing some details, it is not always easy to understand on which samples the analyses were made.

5) Fig. 2C. In the text (lines 145-146) they report “Additionally, young patients are more likely to be negative for estrogen receptor (ER), progesterone receptor (PR), and human epidermal growth factor receptor 2 (HER2). Looking at the graph, I think they intended to say "positive" for HER2 .

6) Supp Fig. 1 show data from the FUSCC study confirming what was observed in the METABRIC study. In panel B the difference between the blue curve (young) and the green curve (intermediate) is not as significant as indicated, since the two curves overlap perfectly. Moreover, in the main figure they say that the worst OS is observed in young women, while in this figure it is in elders. The results are therefore not in confirmation, but in contrast. DFS is not reported.

7) Fig. 3: In panel B they show the most frequently mutated genes by comparing young-intermediate (left) and young-elder (right). TP53 is the most frequently mutated in young women (with p<0.001 as indicated in the text), why does it not appear in the graphs? CDH1 from graph 3B is more significantly mutated in the elder than in the young, but if we look at the graph 3A the mutation percentages are 9% in the elder and 8% in the young... which does not look to be such a marked difference.

8) The analyses to calculate the NES score are done only on the METABRIC cohort, why aren't the other two cohorts, or the integration of the three, also taken into consideration?

9) For the single cell analysis, they focus on the galactose pathway which is one of those that came out of the previous analysis. Why do they value this one?

10) In the in vitro part, the experiments are performed on cell lines showing an enrichment of the galactose pathway and of the stemness signature. In this way, the subsequent experiments in which they demonstrate that galactose increases stemness seems somehow "biased".

11) In line 235 they report “According to the RNA-seq data from 19 cell lines, Hs578t, MDA-MB-231, MDA-MB-436, HCC1395, BT549, HCC1143, and HCC1937 cell lines showed higher enrichment scores of galactose metabolism and stemness signature.” Where did the data on these 19 cell lines come from? Is some data not shown or are they taken from external dataset?

12) In general, M&M and Legends should be better described and the reason/methodological approach for some analyses better described.

The paper is well written and clear.

Some typos, here and there, are to be corrected.

Round 2

Reviewer 3 Report

The author provided a more comprehensive explanation about the cohort they analyzed in the manuscript, adding also a new cohort SCAN-B to validate RNA data.

Still, from what I can appreciate, there is some doubt about how rigorously the analyses have been performed.

-For instance, upon my request, the authors increased the significance between young and elder from * to ** in figure 2..  How? Was it mistaken before? Still, it is strange that the significance between young and intermediate is less then the others, as the curves are the most divergent.

-Further, the authors reported in the cover letter:

“It's worth noting that the FUSCC cohort1, comprising 12,790 operable patients, had a relatively low incidence of relapse or mortality (n=674). This lower frequency made it challenging to discern differences in Disease-Free Survival (DFS) between various age groups. In response to your valuable suggestion, we have included an additional FUSCC cohort (n=4,079). “

In Figure 1 in which they reported the graphical overview of the investigation indicated that FUSCC has 3607 samples…it is difficult to understand why they now talk about an additional cohort of 4079. Did they use this new cohort for DFS KM? IF we sum the number of patients at time 0 is 3607 and not 4079.

This figure is a little bit confusing, a clearer description would be helpful.

-When I asked about CDH1 in Figure 3, the authors talk about a data misalignment. Now, new data appear exactly to show CDH1

Round 3

Reviewer 3 Report

The authors have responded to my concerns.

none